# Microstructure, Mechanical Properties and Fracture Behavior of Micron-Sized TiB_2_/AlZnMgCu(Sc,Zr) Composites Fabricated by Selective Laser Melting

**DOI:** 10.3390/ma16052112

**Published:** 2023-03-06

**Authors:** Peng Yin, Yantao Liu, Zhuoheng Liang, Wei Pan, Shuobing Shao, Yongzhong Zhang

**Affiliations:** 1National Engineering & Technology Research Center for Non-Ferrous Metals Composites, GRINM Group Corporation Limited, Beijing 101407, China; 2Grinm Metal Composites Technology Co., Ltd., Beijing 101407, China; 3General Research Institute for Nonferrous Metals, Beijing 100088, China

**Keywords:** selective laser melting, TiB_2_/AlZnMgCu(Sc,Zr) composites, heterostructure, Al_3_(Sc,Zr), fracture behavior

## Abstract

In this paper, micron-sized TiB_2_/AlZnMgCu(Sc,Zr) composites were fabricated by selective laser melting (SLM) using directly mixed powder. Nearly fully dense (over 99.5%) and crack-free SLM-fabricated TiB_2_/AlZnMgCu(Sc,Zr) composite samples were obtained and its microstructure and mechanical properties were investigated. It is found that the laser absorption rate of powder is improved by introducing micron-sizedTiB_2_ particles, then the energy density required for SLM forming can be reduced, and the densification can finally be improved. Some crystalline TiB_2_ formed a coherent relationship with the matrix, while some broken TiB_2_ particles did not, however, MgZn_2_ and Al_3_(Sc,Zr) can perform as intermediate phases to connect these non-coherent surfaces to aluminum matrix. All these factors lead to an increase in strength of the composite. The SLM-fabricated micron-sized TiB_2_/AlZnMgCu(Sc,Zr) composite finally shows a very high ultimate tensile strength of ~646 MPa and yield strength of ~623 MPa, which are higher than many other aluminum composites fabricated by SLM, while maintaining a relatively good ductility of ~4.5%. The fracture of TiB_2_/AlZnMgCu(Sc,Zr) composite is occurred along the TiB_2_ particles and the bottom of the molten pool. This is due to the concentration of stress from the sharp tip of TiB_2_ particles and the coarse precipitated phase at the bottom of the molten pool. The results show that TiB_2_ plays a positive role in AlZnMgCu alloys fabricated by SLM, but finer TiB_2_ particles should be studied.

## 1. Introduction

In recent years, modern advanced industrial equipment, such as aviation, aerospace, electric power, petrochemical, shipbuilding and so on, is developing towards the direction of complex structure and high performance [1,2]. The requirements for manufacturing technology are increasingly high and the challenges are increasingly severe [3,4]. Selective laser melting (SLM) is a technology that uses a high-power laser to directly melt metal powder into a two-dimensional plane, then superpose layer by layer to form three-dimensional parts [5,6,7,8]. Compared with the traditional process, SLM can directly fabricate nearly fully dense complex parts, which has advantages in shortening production cycle and reducing cost, so it is widely used in military, aviation and aerospace fields [9,10,11].

Al-Zn-Mg-Cu alloys have been widely used in aerospace and automobile industry because of their excellent mechanical properties, extremely high strength-to-weight ratio, excellent thermal working performance, good corrosion resistance and other advantages [12,13]. Studies have shown that Al-Zn-Mg-Cu can form smaller and finer brittle S-phase (Al_2_CuMg) precipitate at a cooling rate of more than 500 °C/h, thus obtaining better aging hardening effect during subsequent heat treatment [14]. The extremely high cooling rate (10^3^~10^6^ K/s) [15] of SLM has great potential for fabricating Al-Zn-Mg-Cu alloys.

However, Al-Zn-Mg-Cu alloys are difficult to weld for complex composition and large solidification temperature range. At the beginning of solidification process, the primary equilibrium phase solidifies preferentially, which leads to the enrichment of solute in the liquid near the solidification interface. Therefore, the temperature of local equilibrium liquid phase changes, resulting in local unstable supercooling state. The primary equilibrium then grows rapidly into columnar grain along the heat flow, leaving behind a thin layer of fluid between branches; after that, the fluid solidifies and shrinks to form cracks parallel to the columnar grain. In contrast, equiaxed grain adapt more easily to the thermal shrinkage strain associated with solidification [15]. Therefore, hot cracks can be effectively eliminated by increasing the proportion of equiaxed grains.

In order to eliminate the hot crack of Al-Zn-Mg-Cu alloys fabricated by SLM, many attempts have been made. John H. Martin et al. introduced Al_3_Zr with basically complete coherent relation into aluminum matrix by adding ZrH_2_. In the process of SLM, Al_3_Zr could be used as heterogeneous nucleation point to increase the proportion of equiaxed grain and refine the grains in the structure of SLM-formed alloy. Additionally, samples with excellent mechanical properties without cracks can be obtained. After heat treatment, the ultimate tensile strength (UTS) is close to 400 MPa and the fracture elongation (FE) is about 4% [15]. Li et al. alloyed Zr element and introduced Si element to increase melt viscosity and used low melting point silica-rich eutectic to fill cracks. In this paper, a new Si- and Zr-modified Al-Zn-Mg-Cu pre-alloying powder was developed for SLM, and the crack-free sample was successfully prepared. The synergistic effect of Si and Zr is helpful to obtain excellent tensile properties (UTS of about 446 MPa and FE of about 6.5% after being heat treated) [16]. Zhou et al. used TiB_2_ instead of Zr to promote the nucleation of fine equiaxed grain. An Al-Zn-Mg-Cu powder co-incorporation with 4 wt.% Si and 2 wt.% TiB_2_ was prepared by high energy ball milling. After addition of Si and TiB_2_, the formability of Al-Zn-Mg-Cu alloy was improved and the cracks were successfully eliminated, but there were a lot of pores. The UTS of Al-Zn-Mg-Cu sample formed by SLM was 556 MPa, and the fracture elongation (FE) was about 4.5% [17]. The presence of Si may generate harmful phase and reduce mechanical properties. For this reason, Zhu et al. introduced Sc with stronger grain refining effect rather than Si to eliminate thermal cracks. After composition design and process optimization, crack-free AlZnMgCu(Sc,Zr) alloy with unprecedented strength-ductility synergism was obtained by SLM. After heat treatment, the UTS was as high as 647 MPa and the FE was 11.6%, which was much higher than the strength of SLM-formed aluminum alloy reported so far [18].

In general, the main idea of eliminating hot cracks is to introduce a grain modifier in Al-Zn-Mg-Cu alloys, for example, adding melt modification elements such as Si, or introducing nucleating elements or ceramic particles to promote grain refinement. In the existing reports, it is generally achieved by pre-alloying powder preparation and externally addition of particle. 

Pre-alloying powder is a method that modified elements or ceramic particles and the matrix composition were synchronously prepared into powder. Its advantage is that the modified elements or in situ reinforced particles could uniformly distribute in the powder. However, the process is complicated and the cost is high. In addition, the amount and shape of the added particles are not easy to control [19]. On the contrary, the method of adding particles externally is to mechanically mix the pre-alloyed matrix powder with the added powder (elemental powder or reinforced particles), then conduct SLM forming. This method can more easily adjust the composition and amount of added material, and effectively reduce the costs [20].

In various alloying elements, Sc and Zr elements have the best effect on eliminating thermal cracks during SLM forming process, because primary Al_3_(Sc,Zr) generated at high cooling rates can significantly increase the proportion of equiaxed grain and refine the grains. Study have shown that the thermal cracks can be completely eliminated when the content of Sc and Zr reaches 0.6% and 0.36%, respectively [21], and its grain refinement effect can also make material both strong and tough. However, it is not economical to further increase the content of Sc and Zr elements to improve the strength due to the high price. Therefore, it is an option to use external ceramic particles to further enhance the strength by improving the laser absorption rate of powder, reducing splatter and refining the grain [19,22,23,24,25]. Besides, due to the high strength and modulus of ceramic particles, its addition can significantly improve the mechanical properties of alloys. However, it is also necessary to consider the binding between ceramic particles and matrix. Cracks may generate at interface between ceramics and matrix because of poorly matched thermal expansion coefficient and some brittle phase at the interface produced by harmful reactions [26,27]. On the contrary, stable and uniform interface reactions can improve mechanical properties [28]. Based on this consideration, TiB_2_ has become an excellent enhancement in AlZnMgCu alloys for its advantages of high melting point, high strength, large modulus, less harmful reaction but good wettability and strong interface bonding with aluminum alloy and so on [29,30,31,32,33].

Therefore, in this paper, an AlZnMgCu(Sc,Zr) powder, which can be used for SLM forming crack-free samples, was prepared by pre-alloying method, and micron-sized TiB_2_ was added by externally added particle method to further enhance the mechanical properties of the alloy. The effect of micron-sized TiB_2_ on the microstructure and properties of AlZnMgCu(Sc,Zr) alloy was explored, and the mechanism was analyzed. This work would lay the foundation for obtaining lower cost and better performance of SLM formed high strength aluminum composites.

## 2. Materials and Methods

### 2.1. Powder Preparation

An AlZnMgCu(Sc,Zr) alloy powder prepared by inert gas atomized was used as the matrix of the composite powder, and its chemical composition was detected by inductively coupled plasma atomic emission spectroscopy (ICP-AES) (Table 1). As can be seen from Table 1, Zn, Sc, Zr and other elements were burned to a certain extent in the process of melting and gas atomization. Micron-sized TiB_2_ particles (1–3 μm, 99%) purchased from Xuzhou Jiechuang New Material Technology Co. Ltd. were mixed into AlZnMgCu(Sc,Zr) alloy powder at 3 wt.%. The ball-to-powder weight ratio was 1.2:1, and the powder was mixed in a mixer for 15 h. 

A laser particle size analyzer (BT-9300S, Bettersize Instruments Ltd, Dandong, China) was used to test the particle size of the prepared composite powder. As the results showed in Appendix A, the particle size of the TiB_2_/AlZnMgCu(Sc,Zr) powder is in the range of 15 μm to 53 μm, with the D10, D50 and D90 of 14.73 μm, 27.37 μm and 43.91 μm, respectively, which met the particle size requirements for SLM forming. 

The surface morphology of the TiB_2_/AlZnMgCu(Sc,Zr) powder was observed by SEM, and the obtained image is shown in Figure 1. According to Figure 1a, the prepared composite powder has good sphericity. In Figure 1b, it is found that most of the fusiform TiB_2_ particles adhered to the surface of the powder, and a small part scattered between the powder. The long diameter of TiB_2_ particles is about 2~6 μm and the short diameter is about 1 μm.

### 2.2. Forming Process

In this experiment, the EP-M250 metal 3D printing system (Beijing Eplus3D Technology Co., Ltd., Beijing, China) was used to form samples, and the laser has a spot diameter of 70 μm and a maximum power of 500 W. The oxygen content in the chamber of 3D printing system is strictly controlled below 100 ppm by filling with Ar gas and the temperature of substrate was preheated and maintained at 140 °C before and during SLM processing. Keep hatch spacing (H) and the layer thickness (D) as constant, while laser power (P) and scanning speed (V) were adjusted to explore their major impact on the quality of the fabricated sample. The SLM process parameters was set as follows: P = 230~260 W, V = 600~1000 mm/s, H = 0.1 mm, D = 0.03 mm and a parallel scanning strategy whose scanning direction rotating is 67° per layer was selected.

Horizontal samples whose size is 60 mm × 10 mm × 11 mm was printed for subsequent exploration, as shown in Appendix A. The long axis of horizontal samples was perpendicular to the building direction (BD), as demonstrated in Figure 2. After several rounds of preliminary process screening, the optimum forming process (OFP) for micron-sized TiB_2_/AlZnMgCu(Sc,Zr) composite was P = 230 W, V = 800 mm/s (A3 sample in Appendix A). Moreover, its energy density (E) was 95 J/mm^3^, calculated by Formula (1) [23]:E = P/(V × H × D)(1)

The forming process of matrix powder AlZnMgCu(Sc,Zr) alloy have also been studied, where the optimum forming process is P = 250 W, V = 760 mm/s and E = 110 J/mm^3^. This part of the work will be described in another article. Compared with AlZnMgCu(Sc,Zr) alloy, the optimum forming energy density decrease after the addition of micron-sized TiB_2_.

The chemical composition of composite powder and formed sample are shown in Table 2. A large amount of Zn element and a small amount of Mg element are vaporized and blown away in the form of soot during the forming process, resulting in the reduction of Zn and Mg content in the fabricated sample. The sum of Ti element and B element in the formed sample is about 3 wt.%, indicating that the micron-sized TiB_2_/AlZnMgCu(Sc,Zr) composite has been successfully prepared. The theoretical density of the formed sample is 2.841 g/cm^3^. 

### 2.3. Experimental Procedures

The density of the composite sample was measured by direct reading solid densitometer (SJ-300G, Shanghai Shuju Instrument Technology Co., Ltd., Shanghai, China), whose working principle is Archimedes drainage method, the ratio of the measured value to the theoretical density was calculated as the densification of the fabricated sample. The horizontal samples were cut into small specimen of 10 mm × 10 mm × 10 mm along the building direction, then the surfaces of the specimen were ground with waterproof abrasive paper and polished to mirror surface by polishing cloth. Internal defects and particle distribution were observed by Axiovert 200 MAT optical microscope (OM). X-ray Computed Tomography (CT), V | tome | x s 240/180, GE Sensing & Inspection Technologies GmbH, Frankfurt, Germany, was performed to highlight the defects within the entire volume of the sample. The following settings were used for the measurements: (i) voltage of 110 kV; (ii) 6.228 μm as resolution of the detector; (iii) 120 min as scanning time. JSM-7900F field emission scanning electron microscope (SEM) equipped with electronic channel contrast (ECC) and electron backscatter diffraction (EBSD, scanning step 0.05 μm) were used to observe and analyze the microstructure of grains. X’Pert Pro MPd polygrainline X-ray diffraction analyzer (XRD) from the Netherlands was used to determine the phase composition of the samples. The working tube voltage was 40 kV, the working current was 40 mA and the scanning step was 0.033°. The microstructures of the interfaces and precipitates were investigated by FEI Tecnai F20 transmission electron microscope (TEM, acceleration voltage 200 kV) equipped with an energy-dispersive X-ray spectrometer (EDX) and a high-angle annular dark field (HAADF).

The TiB_2_/AlZnMgCu(Sc,Zr) composite specimens fabricated by SLM were heat treated with solution treatment at 470 °C for 1 h and then aged at 120 °C for 24 h after quenching in water. The as-printed and heat-treated samples were processed into tensile bar, as shown in Appendix A for mechanical property test, and the room temperature tensile test was carried out by the Quasar10 tensile testing machine. The test standard was GB/T 228.1-2010. Each result was validated at least three times. The samples were cut into tensile pieces along the vertical and horizontal planes respectively to observe the microstructure on the side of the fracture surface after tensile fracture, as shown in Figure 2.

**Figure 2 materials-16-02112-f002:**
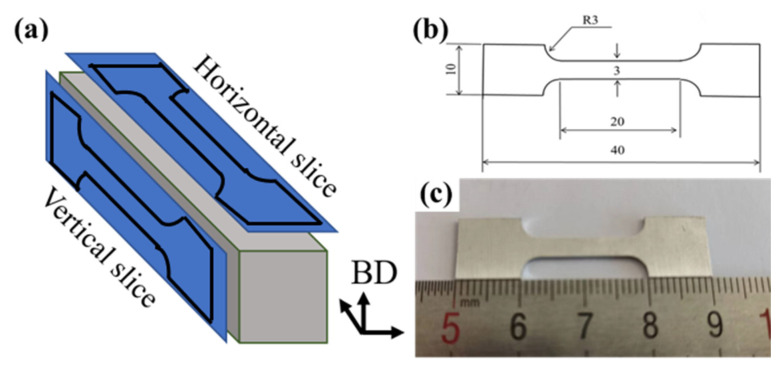
Diagram of tensile plate (**a**) horizontal sample, building direction (BD) and sampling direction, (**b**) schematic diagram (mm) and (**c**) real sample.

## 3. Results and Discussion

### 3.1. Defects and Microstructures Analysis 

The microstructure image of the fabricated sample before and after the addition of TiB_2_ is shown in Figure 3. As shown in Figure 3a,b, there are no cracks in the AlZnMgCu(Sc,Zr) alloy samples obtained by the optimal SLM forming process, but a large number of pores exist, shown as black dots and indicated by the red arrows in the figure. The microstructure of the TiB_2_/AlZnMgCu(Sc,Zr) composite sample is shown in Figure 3c,d. It can be seen from the figure that there are no cracks in the composite material, with only a few pores inside, as shown by the red arrows. The average size of pores in TiB_2_/AlZnMgCu(Sc,Zr) composite sample is about 8 μm, and Appendix A shows the size distribution of pores. The porosity of TiB_2_/AlZnMgCu(Sc,Zr) composite samples is less than that of AlZnMgCu(Sc,Zr) alloy samples, which is also consistent with the increase of densification. Moreover, the light gray spindle-shaped dot phases indicated by the blue arrows in the image are the micron-sized TiB_2_ particles. In the composite material, TiB_2_ particles are uniformly distributed in the matrix, with no obvious agglomeration. Besides, TiB_2_ particles are well combined with the matrix, and no cracks or inclusions were found around the particles. 

The densification of the composites is 99.67%, tested by electronic solid densitometer. Compared with the 98.7% densification of AlZnMgCu(Sc,Zr) sample fabricated by the optimal SLM process in the previous experiment, samples with higher density can be obtained by the addition of micron-sized TiB_2_ by improving the laser absorption rate of powder [17,19,22]. The pore volume of composite samples was measured by industrial CT, and the defect volume ratio is 0.24%, which means that the density of the sample is 99.76%, which is consistent with the density measured by the electronic solid densitometer. The distribution of defects inside the sample is shown in Appendix A. Most of the pores in the samples were spherical, which is consistent with the characteristics of volatile pore. There is also a small number of irregular pores, which is believed to be gaps of some incomplete melting powder. All these pores are uniformly distributed in the matrix. The size and shape of pores have great influence on Young’s modulus, UTS and fracture behavior. There are more concentration points around large pores and irregular pores, where cracks are more likely to start and propagate nearby. The rounder the shape of the pore, the more uniform the stress distribution nearby, and the higher the Young’s modulus and UTS of the material [34]. The surface morphology of the formed sample was observed by using SEM, and EDS scanning was performed for the distribution of the elements. The results are shown in Figure 4, where the bright white fusiform particles in Figure 4a are TiB_2_ particles with irregular shape and some sharp tips, whose particle size is about 5 μm. The general shape of the molten pool can be observed in Figure 4a.

The microstructure of the TiB_2_/AlZnMgCu(Sc,Zr) composite sample after heat treatment is shown in Figure 5. TiB_2_ particles are randomly distributed in the equiaxed fine grain (FG) region and columnar grain (CG) region. The shape and size of TiB_2_ particles are almost unchanged. In addition, there are equiaxial bright white particles distributed in the matrix. Unlike TiB_2_ particles, these equiaxed bright white particles are more concentrated in the equiaxed fine grain region, which are speculated to be precipitated phase after heat treatment.

Figure 6 shows the EBSD inverse polar diagram and grain size distribution of the vertical section of AlZnMgCu(Sc,Zr) alloy sample formed by SLM. It can be seen from the figure that the aluminum alloy has a typical heterostructure with columnar grain and equiaxed fine grain alternately arranged, which make AlZnMgCu(Sc,Zr) alloy samples fabricated by SLM both strong and tough [35,36]. Compared with Figure 6a,b and Figure 6c,d, the grain size does not decrease but rather increases after the addition of micron-sized TiB_2_. The grain size of equiaxed grain is about 1 μm and the columnar grain is about 5 μm in AlZnMgCu(Sc,Zr) alloy, while the equiaxed grain is about 1.5 μm and the columnar grain is about 9 μm in TiB_2_/AlZnMgCu(Sc,Zr) composite. As we can see by comparing Figure 6e,f and Figure 6c,d, after heat treatment, the grain size of TiB_2_/AlZnMgCu(Sc,Zr) composite is further coarsened. The grain size of equiaxed grain increases to about 1.5 μm and the grain size of columnar grain also increases a little.

### 3.2. Precipitated Phase and Interface Analysis

TEM analysis was used to further observe the microstructure of the as-printed composites. Figure 7a,b show the microscopic morphology of the boundary between equiaxed grain and columnar grain in a single molten pool. Comparing the bright field image with the high-angle annular dark field (HAADF) image, it is found that the strip precipitates are evenly distributed along the grain boundaries, and there are some nano-sized round particles in the equiaxed grain boundaries and columnar grain. In addition, the intact micron-sized TiB_2_ particles are embedded between columnar grains. The diameter of equiaxed grain is 0.5~1 μm and the length of columnar grain is about 6~10 μm.

After heat treatment, there are some changes on the microstructure of TiB_2_/AlZnMgCu(Sc,Zr) composite, shown in Figure 7c,d. The most obvious change is that there is no banded precipitate at the grain boundary, which becomes clean. The amount of nano-sized round precipitated particles decrease in the columnar grain region, while increase in the equiaxed grain region. In addition, the equiaxed grains only have limited growth, the average diameter is about 1.3 μm and the columnar grains have coarsened a little.

There are some changes in the precipitated phase after heat treatment observed in STEM, as shown in Figure 7. However, these changes are hard to identify in phase diagrams, as depicted in Appendix A. The reason for this may be the small grain size and low volume fraction of the precipitates, which makes forming discernible peaks in the phase diagram difficult. In order to clarify the phase composition of the precipitated phase, EDS scanning was performed in the equiaxed grain region and the columnar grain region, and the results are shown in Figure 8. In Al matrix, a small amount of Mg and Zn elements are solidly dissolved, and a large amount of Mg and Zn elements are uniformly enriched at the grain boundaries of the equiaxed fine grain region and columnar grain region. Due to the higher energy at the grain boundary, continuous precipitations are formed along the grain boundaries by homogeneous enrichment of Mg and Zn elements [16,18]. In addition, the Mg and Zn elements can be precipitated into a granular MgZn_2_ precipitated phase due to the partial remelting and thermal effects of SLM laser scanning in the later layer. The grains in the equiaxed grain region are generally only subjected to the thermal history of melting and rapid solidification by laser heating, so there is almost no MgZn_2_ precipitated phase within the grains.

Sc and Zr elements are slightly different. Regardless of equiaxed grain zone or columnar grain, Sc and Zr elements do not accumulate along grain boundaries, and most of them are solidly dissolved in Al matrix, while a small amount is precipitated into nano-sized Al_3_(Sc,Zr) particles within grain boundaries. The reasons for this could be as follows: when the bottom of the molten pool contacts with the solidified previous layer, it will cause a very high cooling rate and temperature gradient, which is conducive to the precipitation of primary Al_3_(Sc,Zr), and these primary Al_3_(Sc,Zr) can act as the nucleation center to form ultra-fine grain [37,38]. Columnar grains are grown from the outer edge of equiaxed grains, and may be introduced some primary Al_3_(Sc,Zr) [21]. More likely, a small amount of secondary Al_3_(Sc,Zr) is precipitated similar to Mg and Zn elements.

In addition, by observing the distribution of elements near TiB_2_, it can be found that there is enrichment of Mg, Zn, Sc and Zr elements around some broken TiB_2_ particles in Figure 8a. However, this phenomenon does not exist around the intact large particle TiB_2_ in Figure 8b. The reason for this is that the intact particle TiB_2_ has a good interfacial combination with the aluminum matrix, while the broken TiB_2_ particle has a high free energy similar to the grain boundary, in which is easy to dissolve Mg, Zn, Sc and Zr elements. In addition, the atomic arrangement on the surface of broken TiB_2_ particles is complex and the interface between those surfaces and aluminum matrix is poor. A small amount of precipitated Mg (Zn_1.5_Cu_0.5_) phase can be used as the interphase to improve the overall interface bonding [32,33].

After heat treatment, the distribution of elements changes significantly, as shown in Figure 9. The continuous precipitation at grain boundary disappeared after heat treatment, and the grain boundary became clean and smooth. Mg, Zn and Cu element were uniformly distributed, with only a small number of granular precipitates. In the solution treatment stage, Mg, Zn and Cu element redissolved, while in the aging stage, Mg, Zn and Cu element precipitated in the form of particles at the grain boundary, because of the high energy density of the grain boundary. The number of precipitated particles in the fine grain region was also more than that in the columnar grain region, because there were more grain boundaries in the fine grain region.

Due to the non-equilibrium solidification process caused by extremely high cooling rate in SLM forming process, a large number of Sc and Zr elements were oversaturated in the matrix, then during the heat treatment, these Sc and Zr element elements in oversaturated solid solutions will be precipitated into secondary Al_3_(Sc,Zr) [39,40]. Besides, due to the faster solidification, the amount of Sc and Zr elements in solid solution at the bottom of the molten pool is also higher; moreover, there are more grain boundaries in the fine grain area, which can provide rapid diffusion channels. Therefore, there are more and coarser precipitated particles at the bottom of the molten pool than in the columnar grain region.

Figure 10 presents the interface between micron-sized TiB_2_ particle and Al matrix in SLM-fabricated TiB_2_/AlZnMgCu(Sc,Zr) composite; it can be seen that there is a transition layer of about 5 nm between TiB_2_ particles and Al matrix, and the atoms of the transition layer are misaligned. After calibration of the diffraction pattern from the interface, it is found that there are two orientation relations. They are OR1: (0001) [11−20]TiB_2_ // (−111) [110]Al and OR2: (−110−1) [11−20]TiB_2_ // (002) [110]Al, respectively, as also reported by [19,32,41], indicating that the interface between TiB_2_ particles and Al matrix are well combined.

Crystal structure models of interface between TiB_2_ and aluminum matrix are established to further analyze the interface binding, as shown in Figure 11. The edge-to-edge matching (E2EM) crystallography model is used to calculate the atomic spacing mismatch (*F*), and the calculation Formula (2) [42] is as follows:(2)F=| Dm − Dp | / Dp
where Dm and Dp represent the interatomic distances between the metal matrix and ceramic particles in the close-packed or nearly close-packed directions, respectively.

Figure 11a represents the crystal plane of (−111)Al and (0001)TiB_2_, which have the same structure and similar lattice spacing. The atomic spacing mismatch of the most close-packed and nearly close-packed directions are calculated to be 6.2% and 9.8%, respectively, showing a good coherent relationship. Al_3_(Sc,Zr) has the same grain structure as aluminum matrix, but the lattice spacing is slightly larger. In the same crystal plane pair with TiB_2_, the atomic spacing mismatch of the most close-packed and nearly close-packed directions are calculated to be 4.4% and 4.5%, respectively, showing an excellent coherent relationship, as represented in Figure 11b. Therefore, Al_3_(Sc,Zr) has a better interface binding with TiB_2_ than aluminum matrix dose, so it is more likely to cause enrichment of Sc and Zr elements around TiB_2_.

This phenomenon indicates that TiB_2_ and Sc, Zr element interact with each other and act synergistically in Al matrix composites. On the one hand, it is beneficial to the interface bonding between the reinforcement phase and the matrix, and enhance the strength of the composite material. On the other hand, the evenly distributed precipitation phase Al_3_(Sc,Zr) will be reduced, and the effect of precipitation strengthening will be weaken. These results indicate that there are new considerations in the composition design and strengthening mechanism of Sc, Zr micro-alloyed aluminum matrix composites reinforced with TiB_2_ formed by SLM.

### 3.3. Mechanical Properties

Table 3 shows that the mechanical properties of AlZnMgCu(Sc,Zr) alloys and micron-sized TiB_2_/AlZnMgCu(Sc,Zr) composites after heat treatment with the same process have similar changes: after heat treatment, the tensile strength and yield strength of AlZnMgCu(Sc,Zr) alloys increase about 131 MPa and 155 MPa, respectively, and elongation decreases by about 10.7%. The tensile strength and yield strength of TiB_2_/AlZnMgCu(Sc,Zr) composites increase about 136 MPa and 159 MPa, respectively, and elongation decreases by about 8.5%. The reason for this is that the heat treatment process is mainly regulated by precipitation of Al_3_(Sc,Zr) and MgZn_2_ phases, and the addition of micron-sized TiB_2_ particles has little effect on the heat treatment process. However, the addition of micron-sized TiB_2_ particles improve the mechanical properties. By comparing the mechanical properties of as-printed TiB_2_/AlZnMgCu(Sc,Zr) composites and as-printed AlZnMgCu(Sc,Zr) alloys, the tensile strength has increased by about 7.5%, the yield strength has hardly increased, and the elongation after fracture has decreased by about 19.4%. The properties of heat-treated states are similar, where the tensile strength has increased by about 6.6%, the yield strength has increased by about 3% and the elongation after fracture has decreased by about 15.1%. The heat-treated TiB_2_/AlZnMgCu(Sc,Zr) composite samples obtained by SLM have extremely high strength, where the ultimate tensile strength is about 646.7 MPa and the yield strength is about 623 MPa. In addition, plasticity was preserved, as its elongation after fracture is about 4.5%. Its comprehensive mechanical properties exceed most of the SLM forming aluminum composites reported so far [17,19,20,41,43,44,45,46,47], as shown in Figure 12.

### 3.4. Fracture Behavior

The fracture morphology of as-printed micron-sized TiB_2_/AlZnMgCu(Sc,Zr) composite is shown in Figure 13. It is found that the fracture has a lot of honeycombs structure at low magnification, which is a typical plastic fracture. There are many defects on the surface of the fracture, which are mainly round pores, but no macroscopic cracks are found. Under high magnification electron microscopy, a large number of dimple structures could be observed, and there are some tiny precipitated phases at the bottom of the dimples. In addition, exposed TiB_2_ particles separated from the matrix could be found. The fracture morphology shows that the fracture of micron-sized TiB_2_/AlZnMgCu(Sc,Zr) composite formed by SLM occurs at pores, defects and the interface between the TiB_2_ particles and Al matrix.

The tensile plates were sliced vertically and horizontally, and the microstructure of the side of the fracture were observed by SEM after being grinded and polished; the results are shown in Figure 14. Because the specimens being formed are horizontal, tensile direction (TD) is perpendicular to build-up direction (BD). As can be seen from Figure 14, the direction of growth of crack has little relationship with the shape of molten pool during fracture in as-printed TiB_2_/AlZnMgCu(Sc,Zr) composite. After tensile deformation, the crack passes through the columnar and equiaxed grain regions completely. Moreover, there are TiB_2_ particles and grooves left by debonding of TiB_2_ particles near the fracture. The observation above indicates that the sharp tip of TiB_2_ will cause concentration of stress, which leads to crack propagation.

Heat-treated TiB_2_/AlZnMgCu(Sc,Zr) composite samples were sliced along the vertical direction, and the comparison results with the as-printed state are shown in Figure 15. According to Figure 15b, TiB_2_ particles and grooves left by TiB_2_ particles after debonding also exist in the crack passing through the sample in heat treatment state. The difference is that the crack has a tendency to proceed along the equiaxial fine grain zone, which is the bottom of the molten pool. A possible reason for this is that the large precipitated phase at the bottom of the molten pool, as analyzed in Figure 9, may cause a concentration of stress, so it is easier for cracks to propagate in the fine grain zone [48]. Therefore, in the heat-treated TiB_2_/AlZnMgCu(Sc,Zr) composite specimens, the crack generally expands along the equiaxed grain zone and the interface between TiB_2_ and the matrix.

## 4. Conclusions

In this paper, the micron-sized TiB_2_/AlZnMgCu(Sc,Zr) composites fabricated by SLM are investigated. The differences of micron-sized TiB_2_/AlZnMgCu(Sc,Zr) composite and AlZnMgCu(Sc,Zr) alloy samples in terms of macroscopic defects, microstructure, interface, mechanical properties and fracture behavior are compared between the as-printed and heat-treated samples. The main conclusions are as follows:
The crack-free micron-sized TiB_2_/AlZnMgCu(Sc,Zr) composite with good surface quality and high densification of 99.76% have been successfully prepared by SLM. Micron-sized TiB_2_ particles are distributed uniformly in the matrix. The addition of TiB_2_ leads to an increase of densification, but has little influence on the microstructure, phase composition and heat-treatment process.The complete micron-sized TiB_2_ particle surface has a good interface with the aluminum matrix and the interface is clean and smooth, with no other intermediate phase. A large number of alloy-elements are enriched near the irregular surface of these broken TiB_2_, for relatively high surface free energy, similar to grain boundaries. Due to the poor interfacial bonding between these parts of the surface and the aluminum matrix, the Mg (Zn_1.5_Cu_0.5_) phase can form as the interphase to bond the matrix. The reason for this is that its lattice parameter at a specific crystal plane is between the TiB_2_ and aluminum matrix.The addition of micron-sized TiB_2_ particles has a beneficial effect on the mechanical properties of AlZnMgCu(Sc,Zr) samples, and the increase is about 6%. The main reason for the increase is the increase of densification due to the increase of the powder laser absorption rate. In contrast, due to the introduction of brittle ceramic particles and its stress concentration effect, the ductility has decreases by about 15%. The ultimate tensile strength and yield strength of the composite are about 646 MPa and 623 MPa, respectively. Moreover, the composite has some plasticity and elongation after fracture, which is about 4.5%.The fracture of the micron-sized TiB_2_/AlZnMgCu(Sc,Zr) composite is due to the stress concentration of the sharp tips of TiB_2_. In addition, the large particles precipitated after heat treatment are mainly concentrated in the fine grain zone at the bottom of the molten pool, and these coarse precipitated phases can also easily cause propagation of cracks.

To sum up, the introduction of micron-sized TiB_2_ particles by the external method do have a positive effect, such as improving the densification and slightly improving the mechanical properties. However, there is space for further improvement. The reason for this is that the micron-sized TiB_2_ particles introduced by the external method are too large to achieve the effect of fine grain strengthening, and the effect of Orowan strengthening and load bearing is also minimal due to insufficient additions. Besides, most of the particles added are irregular in shape, which may cause stress concentration. Therefore, the use of nano-sized particles TiB_2_ should be the focus of subsequent experiments, and the synergistic effect of TiB_2_ and Sc, Zr micro-alloyed Al matrix composites can also be further studied.

## Figures and Tables

**Figure 1 materials-16-02112-f001:**
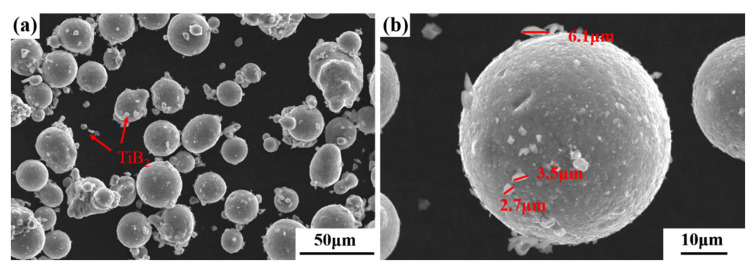
SEM morphology of 3 wt.% micron-sized TiB_2_/AlZnMgCu(Sc,Zr) composite powder (**a**) in relatively low magnification and (**b**) in relatively high magnification.

**Figure 3 materials-16-02112-f003:**
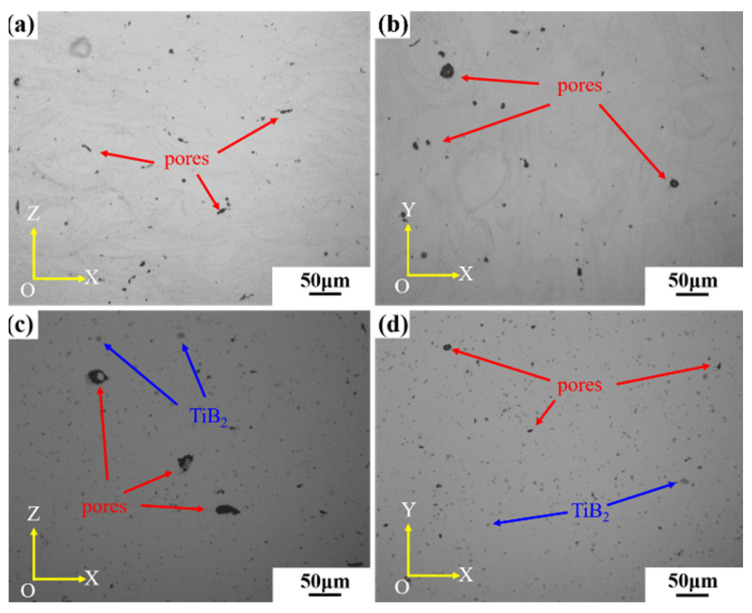
Metallography of SLM formed as-printed specimen: (**a**) vertical plane of AlZnMgCu(Sc,Zr) alloy, (**b**) horizontal plane of AlZnMgCu(Sc,Zr) alloy, (**c**) horizontal plane of TiB_2_/AlZnMgCu(Sc,Zr) composite and (**d**) horizontal plane of TiB_2_/AlZnMgCu(Sc,Zr) composite.

**Figure 4 materials-16-02112-f004:**
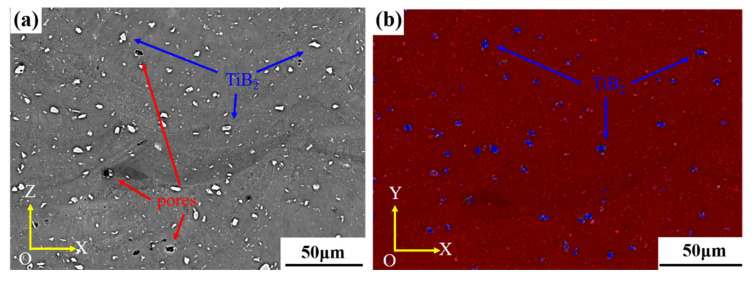
The microstructure of as-printed TiB_2_/AlZnMgCu(Sc,Zr) composite samples: (**a**) SEM and (**b**) EDS planar scanning.

**Figure 5 materials-16-02112-f005:**
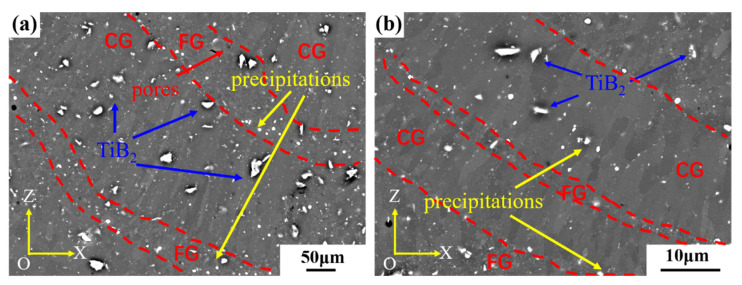
The microstructure of heat-treated TiB_2_/AlZnMgCu(Sc,Zr) composite samples (**a**) in relatively low magnification and (**b**) in relatively high magnification.

**Figure 6 materials-16-02112-f006:**
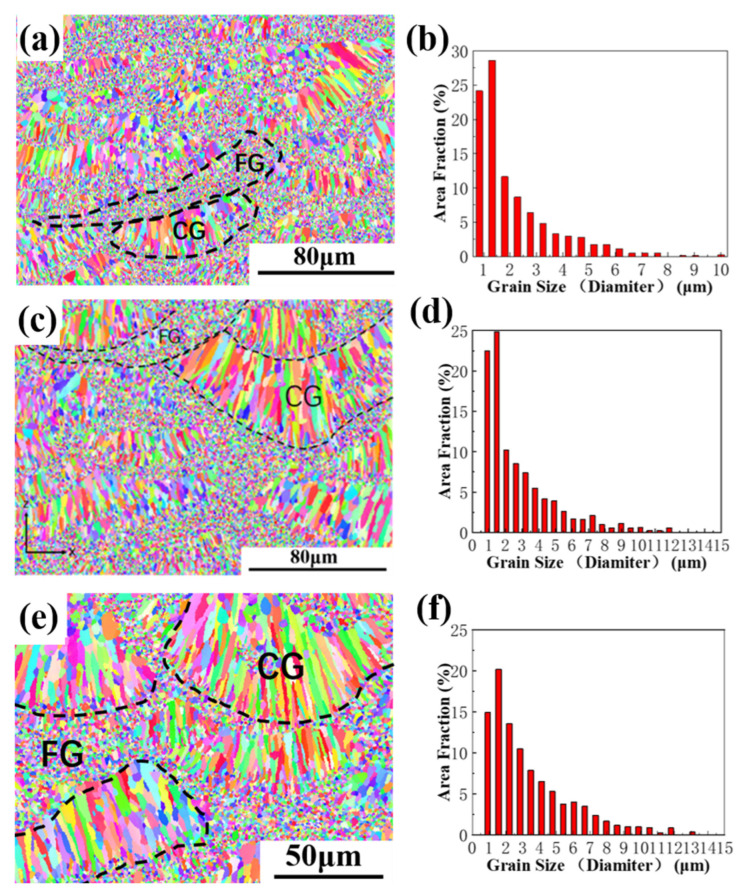
EBSD analysis and grain size distribution in vertical plane of SLM forming specimen (**a**,**b**) as-printed AlZnMgCu(Sc,Zr) alloy, (**c**,**d**) as-printed TiB_2_/AlZnMgCu(Sc,Zr) composite and (**e**,**f**) heat-treated TiB_2_/AlZnMgCu(Sc,Zr) composite.

**Figure 7 materials-16-02112-f007:**
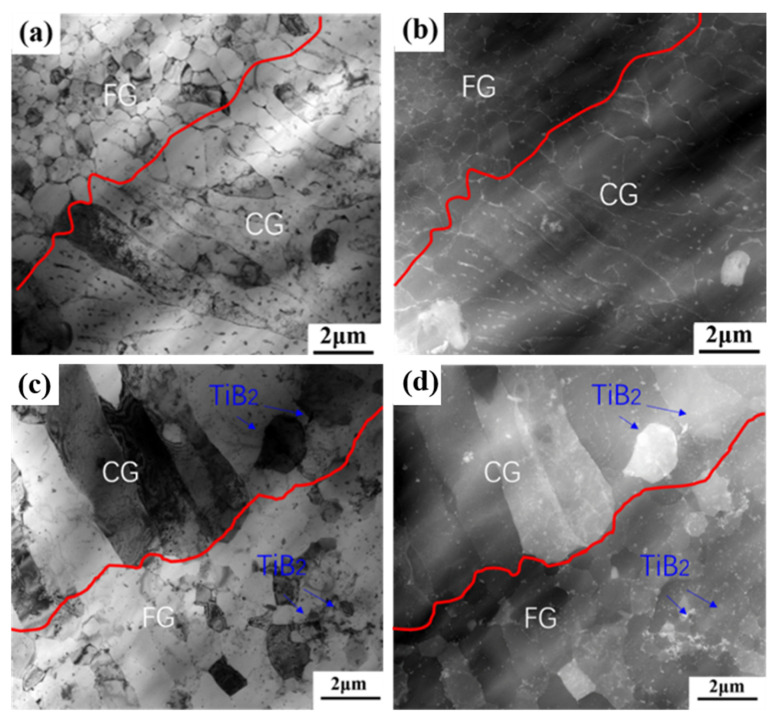
Morphological characteristics of equiaxed and columnar grain in TiB_2_/AlZnMgCu(Sc,Zr) composite under STEM, as-printed state: (**a**) bright filed (**b**) HAADF and heat-treated state: (**c**) bright filed and (**d**) HAADF.

**Figure 8 materials-16-02112-f008:**
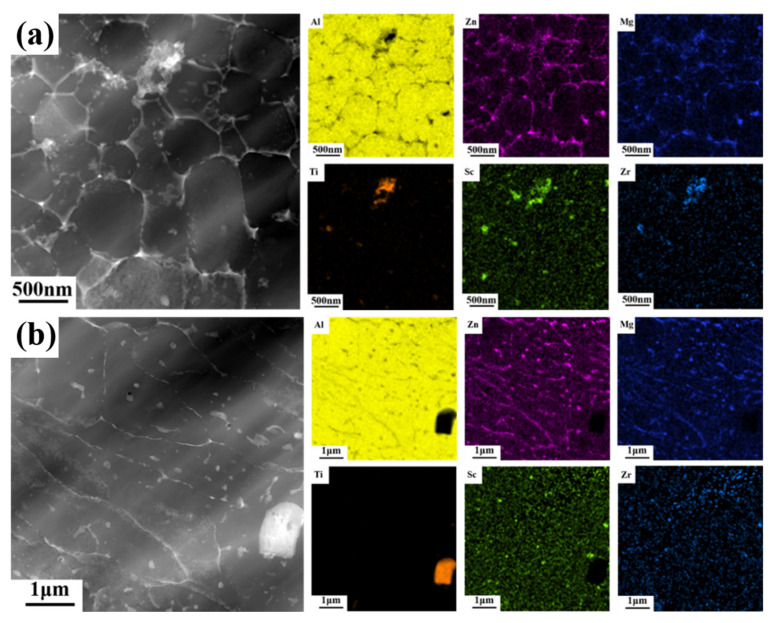
STEM HAADF images of SLM-fabricated TiB_2_/AlZnMgCu(Sc,Zr) composite as-printed samples and corresponding EDX mapping of the solute elements (Al, Zn, Mg, Ti, Sc, Zr) (**a**) in the fine grain region and (**b**) in the columnar grain region.

**Figure 9 materials-16-02112-f009:**
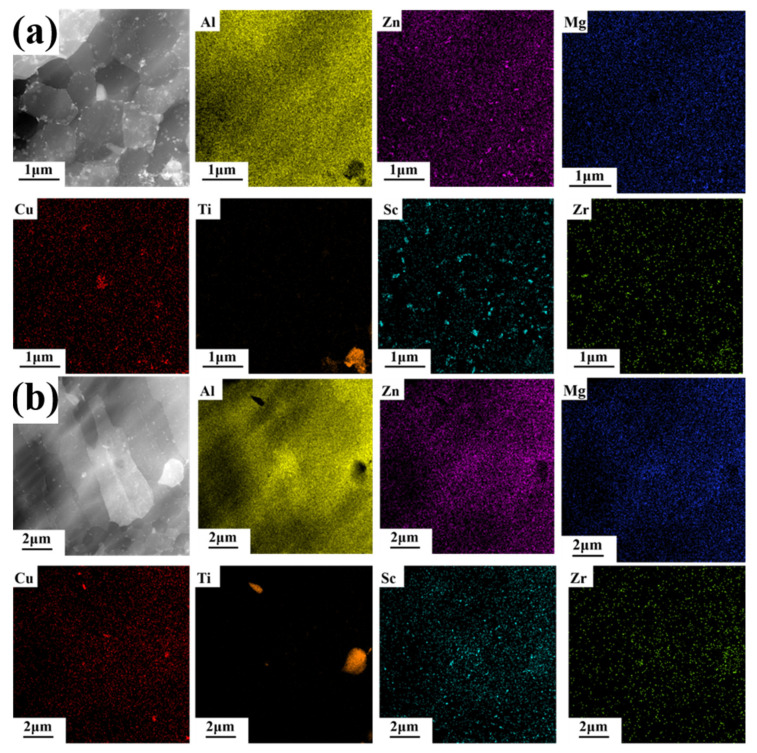
STEM HAADF images of SLM-fabricated TiB_2_/AlZnMgCu(Sc,Zr) composite heat-treated samples and corresponding EDX mapping of the solute elements (Al, Zn, Mg, Cu, Ti, Sc, Zr) (**a**) in the fine grain region and (**b**) in the columnar grain region.

**Figure 10 materials-16-02112-f010:**
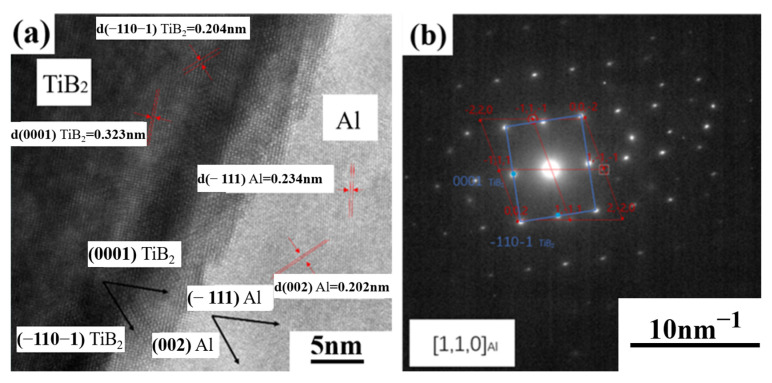
(**a**) HRSTEM image of the interface between micron-TiB_2_ particle and Al matrix in SLM-fabricated TiB_2_/AlZnMgCu(Sc,Zr) composite and (**b**) indexing of the SAED patterns, showing relative orientation relations (ORs).

**Figure 11 materials-16-02112-f011:**
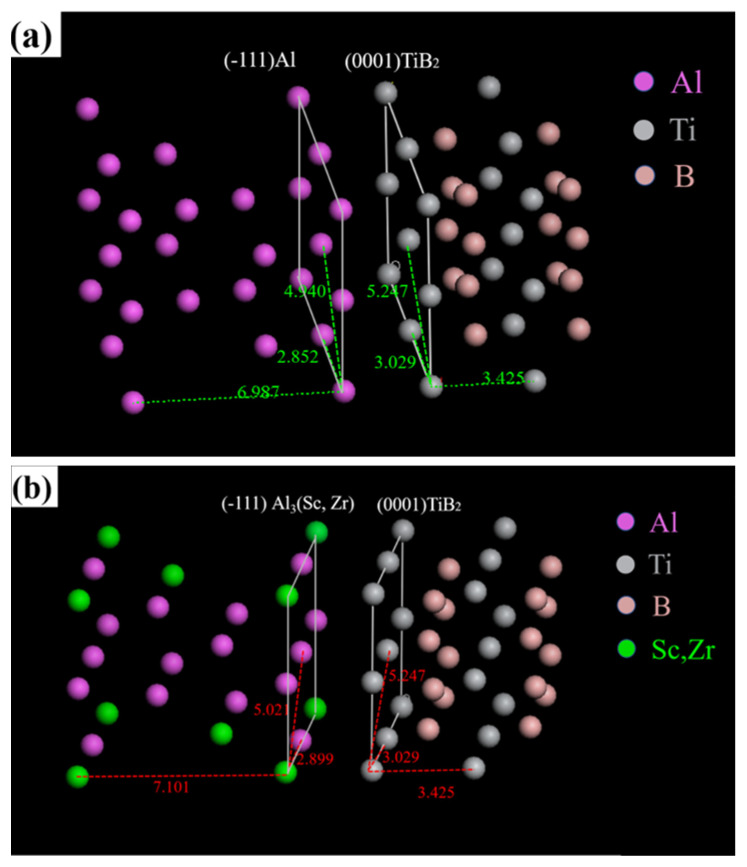
Crystal structure models of interface between TiB_2_ and AlZnMgCu(Sc,Zr) matrix, (**a**) (−111)Al and (0001)TiB_2_, (**b**) (−111)Al_3_(Sc,Zr) and (0001)TiB_2_.

**Figure 12 materials-16-02112-f012:**
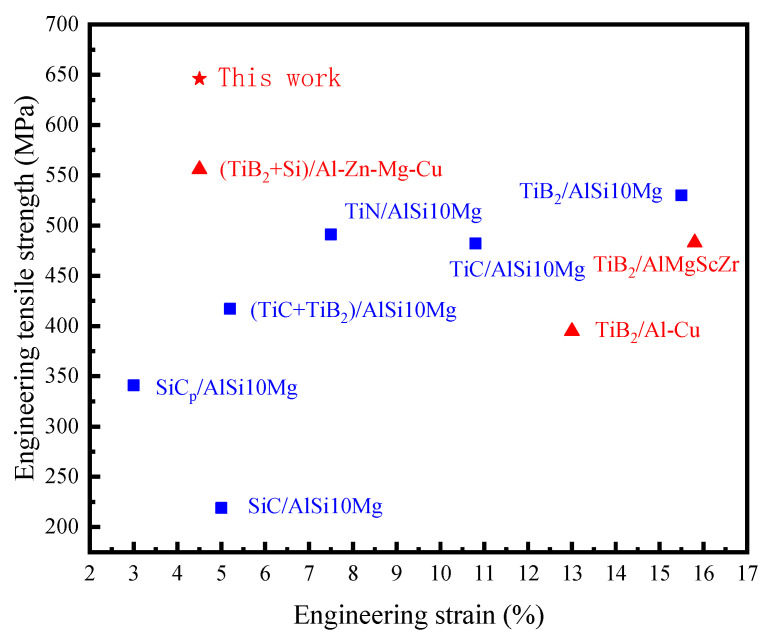
Comparison of mechanical properties of aluminum matrix composites formed by SLM [17,19,20,41,43,44,45,46,47].

**Figure 13 materials-16-02112-f013:**
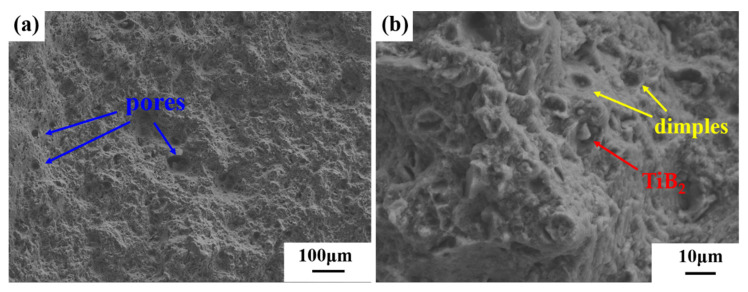
Fracture surfaces of SLM-fabricated TiB_2_/AlZnMgCu(Sc,Zr) composite as-printed sample (**a**) in relatively low magnification and (**b**) in relatively high magnification show dimples.

**Figure 14 materials-16-02112-f014:**
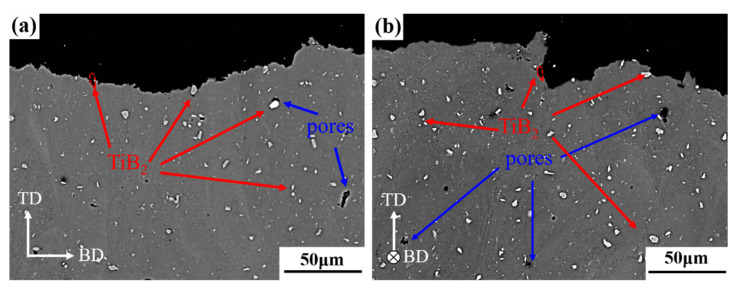
Fracture side surfaces of SLM-fabricated TiB_2_/AlZnMgCu(Sc,Zr) composite as-printed sample in relatively low magnification: (**a**) vertical plane and (**b**) horizontal plane.

**Figure 15 materials-16-02112-f015:**
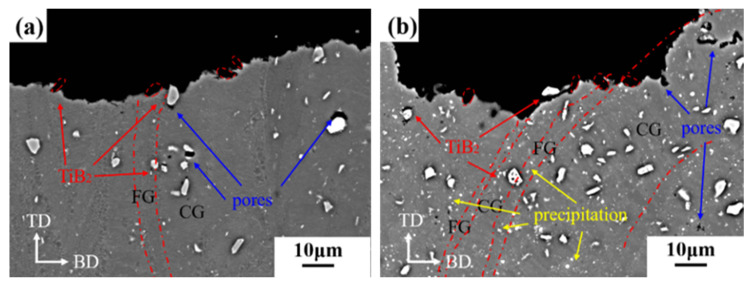
Fracture side surfaces of SLM-fabricated TiB_2_/AlZnMgCu(Sc,Zr) composite vertical plane in relatively high magnification: (**a**) as-printed sample and (**b**) heat-treated sample.

**Table 1 materials-16-02112-t001:** The composition of AlZnMgCu(Sc,Zr) powder.

Elements/wt.%	Zn	Mg	Cu	Sc	Zr	Al
Design content	8.5	3	1	0.8	0.6	Bal.
Powder content	8.28	2.96	1.02	0.66	0.39	Bal.

**Table 2 materials-16-02112-t002:** The composition of micron-sized TiB_2_/AlZnMgCu(Sc,Zr) powder and the formed sample.

Elements/wt.%	Zn	Mg	Cu	Ti	B	Sc	Zr	Al
Powder	8.44	3.12	0.98	1.34	0.52	0.66	0.40	Bal.
Sample	6.24	2.60	1.01	2.05	0.85	0.68	0.41	Bal.

**Table 3 materials-16-02112-t003:** Comparison of mechanical properties of AlZnMgCu(Sc,Zr) alloy and TiB_2_/AlZnMgCu(Sc,Zr) composite before and after heat treatment.

Samples		UTS (MPa)	YS (MPa)	Elongation (%)
AlZnMgCu(Sc,Zr)	As-printed	475.3 ± 0.8	448.7 ± 4.0	16.0 ± 0.5
	Heat treated	606.7 ± 14.5	603.7 ± 18.0	5.3 ± 1.5
TiB_2_/AlZnMgCu(Sc,Zr)	As-printed	510.8 ± 16.5	452.3 ± 17.1	13.0 ± 1.3
	Heat treated	646.7 ± 9.0	623.1 ± 22.6	4.5 ± 2.3

## Data Availability

Date are contained within the article.

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
