# Peer review of "Microstructure, Mechanical Properties and Fracture Behavior of Micron-Sized TiB2/AlZnMgCu(Sc,Zr) Composites Fabricated by Selective Laser Melting"

_materials, 2023, doi:10.3390/ma16052112_

Round 1

Reviewer 1 Report

1. Fig. 1 The caption must include more description about the labels D90, D50 and D10.

2. The labels in Fig. 2 are not easily readable.

3. What is meant by the term "horizontal sample". It has to be introduced with reference to the coordinate system of the SLM device.

4. The full form of acronym BD in figure 5 has to be presented in the caption. 

5. The labels in Fig. 6, 8 and the scale bar in Fig. 7 are not easily readable.

6. It is recommended to combine some figures to reduce the total number of figures. 

7. The keywords list can be expanded using the terminology related to structure and property/behaviour of the materials studied by SLM.

Reviewer 2 Report

General comments

In this research, the microstructure and mechanical properties, including fracture behavior, of additive manufacture TiB2/AlZnMgCu(Sc,Zr) composite material manufacture by selective laser melting (SLM) was studied. This work is interesting, novel and well written and it is of great relevant for the readers of Materials. The results are well presented and discussed. Yet, there are too many figures and tables in this manuscript (26 figures and 3 tables). Therefore, it is recommended to move some of the figure to a Supplementary Materials file. Thus, I recommend publishing this contribution in Materials, MDPI, after minor revision, as explained below.

Title

(1) Title, Page 1: “Microstructure”, please correct to “Microstructure,”   

Materials and Methods

(2) Page 4, Figure 2: You need to improve the red letters quality.

(3) Page 7, Figure 4: Please add units of length.

(4) Page 6, lines 185-186: Axiovert 200 MAT optical microscope (OM)” should be changed to “Axiovert 200 MAT optical light microscope (LM)”.

Results and Discussion

(5) Title: Please change the change to capital letter “D” of the word discussion. Should be “Results and Discussion”.

(6) Page 8, Figure 6: How was the samples ground and polished? In the Materials and Methods part please add an explanation about the preparation of the metallographic samples. In addition you need to improve and enlarge the font in Figure 6.

(7) Page 8, line 224: “The pore volume of composite samples was measured by industrial CT”. Please add more details concerning the CT instrument and testing parameters.

(8) Page 8, line 228: “with diameters between 0.3 and 0.7 um” – You need to correct to 0.7 μm.

(9) Page 9, line 236: “whose particle size is about 5μm” – You need to add space after the number and before the units, should be “5 μm”.

(10) Page 9, Figure 8: You need to improve and enlarge the font in the figure.

(11) Page 11, Figure 8: “TEBSD analysis and grain size distribution in vertical plane of S

LM forming specimen (a) 267 (b) as-printed AlZnMgCu(Sc,Zr) alloy, (c) (d) as-printed TiB2/AlZnMgCu(Sc,Zr) composite and (e) 268 (f) heat treated TiB2/AlZnMgCu(Sc,Zr) composite” – This should be Fig. 10.

(12) Page 11, line 278: MgZn2 should be corrected to MgZn2 (the number 2 should be subscript).

(13) Pages 13-16, lines 329, 331 and 395: Change TiB2 to TiB2 (2 should be subscript).

Conclusions

(14) Page 21, line 513: Change TiB2 to TiB2 (2 should be subscript).

Reviewer 3 Report

Review report on the manuscript Number: materials-2238846, submitted to the Materials.

 Title: Microstructure, mechanical properties and fracture behavior of micron-sized TiB2/AlZnMgCu(Sc,Zr) composites fabricated by selective laser melting

 In the paper, the authors presented the results of research on the microstructure and selected properties of composites obtained by selective laser melting. The composite produced by them has the composition of TiB2/AlZnMgCu(Sc,Zr).

The title of the manuscript reflects its content, but this is insufficient to consider the paper for publication in Materials. Neither the abstract nor the results and discussion convincingly show the novelty of the issues discussed.

General remarks

Q1) The authors have cited the paper [18] were excellent combination of strength (647 MPa) and ductility (11,6%) was obtained for the SLM-produced AlZnMgCuScZr composite. Why these results are not included into the Figure 23?

Q2) What was the main purpose of undertaking the research and presenting its results, since the achievements of the authors of the peer-reviewed manuscript are not better?

Structure of manuscript

Q3) The manuscript is too long and does not have the character of a scientific research article, more like a report of some research project. The experimental studies are well designed, but presented in excessive detail.

Q4) Chapter 3. Results and discussion – The discussion should be not only an attempt to put forward hypothesis confirming the obtained result, but also a reference to the literature, which is rather lacking.

Q5) Figure 3, Figure 4, Figure 5 - rather redundant large pictures showing trivial things

Q6) Figure 7. - "y" axis scale illegible

Q7) Figure 6, Figure 8, Figure 9, Figure 24; Figure 25 -  markings on the microscope pictures are completely unreadable.

Author Response

Dear Reviewer:

Thank you for your comments concerning our manuscript entitled “Microstructure, mechanical properties and fracture behavior of micron-sized TiB2/AlZnMgCu(Sc,Zr) composites fabricated by selective laser melting” (ID: materials-2238846). Those comments are all valuable and very helpful for revising and improving our paper, as well as the important guiding significance to our researches. We have studied comments carefully and have made correction which we hope meet with approval.

According to your suggestions, some trivial things had been reduced in the article, figures were also re-selected, combined and rearrange, the extra figures were put into the Supplementary Materials file. It was also scrutinized in detail. However, due to my limited level and lack of experience, there may be still a lot of room for improvement. Looking forward to your further comments and guidance.

Revised portion are marked in yellow in the revised manuscript.

The main corrections in the paper and the responds to the reviewer’s comments are as following:

Point 1) The authors have cited the paper [18] were excellent combination of strength (647 MPa) and ductility (11.6%) was obtained for the SLM-produced AlZnMgCuScZr composite. Why these results are not included into the Figure 23?

Response 1: Thanks for your reminding, your question is indeed critical. The designed alloy composition of Al-6.78Zn-2.53-Mg-1.94Cu-0.46Sc-0.23Zr-0.09Si-0.07Fe in paper [18] is indeed an unprecedented achievement. The composition of our matrix is similar to their work, Al-6.41Zn-2.38Mg-0.9Cu-0.65Sc-0.37Zr, but the performance of the matrix is not as good as theirs. Therefore, we hope to further improve the mechanical properties by introducing TiB2 particles. Although the improvement is not too much, which may be limited by the addition amount and particle size. After addition of TiB2 particles, the strength of TiB2/AlZnMgCu(Sc,Zr) composites can be close to this level. However, there is still a gap in plasticity. Once ceramic particles are added to the Al alloy, cracks are more likely to occur during the SLM forming process, and the materials become harder and more brittle. Therefore, in addition to comparing the mechanical properties of AlZnMgCuScZr before and after the addition of TiB2 particles, this work also needs to compare with other composites, as shown in Figure 12, because the challenges faced  and the problems studied in these works are more similar.

Point 2) What was the main purpose of undertaking the research and presenting its results, since the achievements of the authors of the peer-reviewed manuscript are not better?

Response 2: In order to prepare AlZnMgCu alloy suitable for SLM forming as in paper [18], which is not only crack-free, but also strong and tough, it is necessary to design the composition of complex elements. However, it takes time and money to explore the available combinations of compositions by pre-alloying powder and then SLM forming. We have designed a component with good performance as mentioned in Response 1, but there is still space for improvement in mechanical properties. Therefore, we hope to further enhance the mechanical properties by adding TiB2 particles externally, which is a convenient and efficient method. In the literature reported so far, the combination of AlZnMgCu alloy and Sc, Zr elements and the combination of AlZnMgCu alloy and TiB2 particle have been well explored in SLM forming, but the combination of the three is relatively rare. The interaction between TiB2 and AlZnMgCu has also been thoroughly studied. However, there are still relatively few reports on the effect of addition of TiB2 on the precipitation behavior of Sc, Zr element. By studying these problems, we can also find some limitations in AMCs formed by SLM, such as stress concentration caused by large particles and irregular shapes, which will lay the foundation for further exploration of better TiB2 addition methods, such as changing the amount of addition, using finer particles, or even using in-situ reinforcement. We think the present work has reference significance for our follow-up works. Although the mechanical properties of our work are not as excellent as reported in paper [18], the composite materials may have their outstanding aspects in hardness, elastic modulus, thermal conductivity and wear resistance, which are the areas we plan to explore in the future.

Thank you for your questions and comments again. Hopefully, this Response will clarify our main purpose of undertaking the research and presenting its result, and we are looking forward to hearing your further questions and comments.

Point 3) The manuscript is too long and does not have the character of a scientific research article, more like a report of some research project. The experimental studies are well designed, but presented in excessive detail.

Response 3: We are sorry for being not very experienced enough in writing scientific research article. And we also lack of experience in choosing which works to present and tend to present excessive detail. These are the things we need to learn. According to your suggestion, we re-selected, combined and rearranged the pictures, and put the extra pictures into the Supplementary Materials file, reducing the number of figures in the manuscript to 15. In addition, we also deleted some descriptions that are too detailed and reduced the paper to 20 pages. If necessary, we can further simplify the details of the experimental process and the description of the figures in the results. Thank you for your valuable advice, and we are looking forward to hearing your further questions and comments.

Point 4) Chapter 3. Results and discussion– The discussion should be not only an attempt to put forward hypothesis confirming the obtained result, but also a reference to the literature, which is rather lacking.

Response 4: We are sorry for this problem. Due to our lack of experience, after introducing the results of some literature in Introduction, we directly used the conclusions of these literature in Results and Discussion without re-citing them. So, the conclusions we referenced were re-cited and some added literature  were also referenced, all these added citations were highlighted in yellow. Thank you for your valuable advice, and we are looking forward to hearing your further questions and comments.

Point 5) Figure 3, Figure 4, Figure 5 - rather redundant large pictures showing trivial things

Response 5: Thanks for your valuable advice, we put these extra pictures into the Supplementary Materials file.

Point 6) Figure 7. - "y" axis scale illegible

Response 6: We are sorry that we did not carefully review the manuscript. Thank you for reminding. We revised this figure by enlarging the font and put it in Supplementary materials as Supplementary Figure 5.

Supplementary Figure 5. Distribution of micropores in TiB2/AlZnMgCu(Sc,Zr) composite samples

Point 7) Figure 6, Figure 8, Figure 9, Figure 24; Figure 25 -markings on the microscope pictures are completely unreadable.

Response 7: We are sorry that we did not carefully review the manuscript. Thank you for pointing out these problems to us. We revised these Figures, improved the markings quality and enlarged the font.

Figure 3. Metallography of SLM formed as-printed specimen (a) vertical plane of AlZnMgCu(Sc,Zr) alloy, (b) horizontal plane of AlZnMgCu(Sc,Zr) alloy, (c) horizontal plane of TiB2/AlZnMgCu(Sc,Zr) composite and (d) horizontal plane of TiB2/AlZnMgCu(Sc,Zr) composite

Figure 4. The microstructure of as-printed TiB2/AlZnMgCu(Sc,Zr) composite samples (a) SEM and (b) EDS planar scanning

Figure 5. The microstructure of heat-treated TiB2/AlZnMgCu(Sc,Zr) composite samples (a) in relatively low magnification, and (b) in relatively high magnification

Figure 13. Fracture surfaces of SLM-fabricated TiB2/AlZnMgCu(Sc,Zr) composite as-printed sample, (a) in relatively low magnification, and (b) in relatively high magnification show dimples

Figure 14. Fracture side surfaces of SLM-fabricated TiB2/AlZnMgCu(Sc, Zr) composite as-printed sample in relatively low magnification (a) vertical plane and (b) horizontal plane

We tried our best to improve the manuscript and made some changes in the manuscript. These changes will not influence the content and framework of the paper.

We appreciate for your warm work earnestly, and hope that the correction will meet with approval.

We look forward to hearing from you regarding our submission. We would be glad to respond to any further questions and comments that you may have.

Once again, thank you very much for your comments and suggestion. 

Best regards,

Mr. Peng yin

Reviewer 4 Report

The manuscript "Microstructuremechanical properties and fracture behavior of 2 micron-sized TiB2/AlZnMgCu(Sc,Zr) composites fabricated by selective laser melting" deals with some interesting aspects of a useful approach in selective laser melting of a composite. In this work, micron-sized TiB2/AlZnMgCu(Sc,Zr) composites were fabricated by selective laser melting (SLM) using directly mixed powder. Nearly fully dense (over 99.5%) and crack-free SLM-fabricated TiB2/AlZnMgCu(Sc,Zr) composite samples were obtained and its microstructure and mechanical properties were investigated. The results showed that the laser absorption rate of powder was improved by introducing micron-sized TiB2, then the energy density required for SLM forming can be reduced, and finally the densification can be improved. The authors have nicely represented the work with interesting data and results. The microstructure analysis results are very interesting as well. I recommend this manuscript for publication after following mandatory revisions.

1.     In several places, authors have mentioned that a crack-free micron-sized TiB2/AlZnMgCu(Sc,Zr) composite has been successfully prepared by SLM. But, almost nothing is mentioned about the mechanisms of crack formation and why the presence of this particle has mitigated the formation of micro-cracks. Authors are highly recommended to consult following manuscripts for a short discussion on crack-formation issue. Although they are not directly related to your system, they have some useful discussions on the topic:

-        Microstructure and mechanical properties of ultrasonic spot welding TiNi/Ti6Al4V dissimilar materials using pure Al coating. Journal of Manufacturing Processes, 64, 2021, 473-480. doi: https://doi.org/10.1016/j.jmapro.2021.02.009

-        Investigation of welding crack in micro laser welded NiTiNb shape memory alloy and Ti6Al4V alloy dissimilar metals joints. Optics & Laser Technology, 91, 2017, 197-202. doi: https://doi.org/10.1016/j.optlastec.2016.12.028

2.     Please provide a quantitative analysis of void size/size distribution.

3.     How many times have you performed mechanical tests?

4.     Authors have mentioned “Due to the poor interfacial bonding between these part of the surface and the aluminum matrix, Mg (Zn1.5Cu0.5) phase or Al3(Sc,Zr) phase may be formed as the interphase to bond the matrix. The reason is that their lattice parameter at specific crystal plane is between TiB2 and aluminum matrix”.  I really do not think it is a good idea to talk about probabilities in conclusion. I suggest bring definite and certain findings in the conclusion. You can move this part to the discussion.

5.     Again in paragraph 4: “The main reason for the increase may be the increase of densification due to the increase of powder laser absorption rate”. I do not like this uncertainty in the conclusion.

6.     Is 6% increase really a meaningful increase? Could that be a scattering in data?

7.     The porosity is a very key issue in your paper. Yet, the discussion on the implications of porosity for the mechanical properties of your samples is very limited. I recommend you consult the following paper and add some discussion using this paper:

-        Statistical effects of pore features on mechanical properties and fracture behaviors of heterogeneous random porous materials by phase-field modeling. International Journal of Solids and Structures, 264, 2023, 112098. doi: https://doi.org/10.1016/j.ijsolstr.2022.112098

8.      A revision of English is highly recommended

Author Response

Dear Reviewer:

Thank you for your comments concerning our manuscript entitled “Microstructure, mechanical properties and fracture behavior of micron-sized TiB2/AlZnMgCu(Sc,Zr) composites fabricated by selective laser melting” (ID: materials-2238846). Those comments are all valuable and very helpful for revising and improving our paper, as well as the important guiding significance to our researches. We have studied comments carefully and have made correction which we hope meet with approval. Revised portion are marked in yellow in the paper. We also re-selected, combined and rearranged the pictures, and put the extra pictures into the Supplementary Materials file, reducing the number of figures in the manuscript to 15. The revised manuscript was uploaded in attathment.

The main corrections in the paper and the responds to the reviewer’s comments are as following:

Point 1) In several places, authors have mentioned that a crack-free micron-sized TiB2/AlZnMgCu(Sc,Zr) composite has been successfully prepared by SLM. But almost nothing is mentioned about the mechanisms of crack formation and why the presence of this particle has mitigated the formation of micro-cracks. Authors are highly recommended to consult following manuscripts for a short discussion on crack-formation issue. Although they are not directly related to your system, they have some useful discussions on the topic:

-Microstructure and mechanical properties of ultrasonic spot welding TiNi/Ti6Al4V dissimilar materials using pure Al coating. Journal of Manufacturing Processes, 64, 2021, 473-480. doi: https://doi.org/10.1016/j.jmapro.2021.02.009

-Investigation of welding crack in micro laser welded NiTiNb shape memory alloy and Ti6Al4V alloy dissimilar metals joints. Optics & Laser Technology, 91, 2017, 197-202. doi: https://doi.org/10.1016/j.optlastec.2016.12.028

Response 1: The formation mechanism and elimination principle of cracks in SLM forming AlZnMgCu are described in Introduction Paragraph 3. The thermal cracks can be significantly eliminated by refining the grains, mainly by adding Sc and Zr elements to form the primary Al3(Sc,Zr) as heterogeneous nucleation. Therefore, through the alloy composition design, the matrix AlZnMgCu(Sc,Zr) samples fabricated by SLM used in this experiment is crack-free. Some nano-sized ceramic particles also have grain refining effect, but the particle size of TiB2 added in this experiment is similar to that of grain size. Combined with the grain size analysis before and after addition, there is no obvious refining effect. Therefore, it is not considered that the addition of micron-sized TiB2 can significantly eliminate cracks, only to improve the process and enhance the strength in this experiment. For clearer explanation, we added some discussion related to cracks and interface bonding in the Introduction Paragraph 6,line 97 and line107, the added paragraph and consulted manuscripts were as below.

Sc and Zr elements have the best effect on eliminating thermal cracks during SLM forming process, because primary Al3(Sc,Zr) generated at high cooling rates can significantly in-crease the proportion of equiaxed grain and refine the grains. Study have shown that the thermal cracks can be completely eliminated when the content of Sc and Zr reaches 0.6% and 0.36%, respectively [21].

Besides, due to the high strength and modulus of ceramic particles, its addition to the alloy can significantly improve the mechanical properties. However, it is also necessary to consider the binding condition of the matrix. Cracks may generate at interface between ceramics and matrix because of poorly matched thermal expansion coefficient and some brittle phase at the interface produced by harmful reactions [26,27]. On the contrary, stable and uniform interface reactions can improve mechanical properties [28]

Pan, W.; Zhai, Z.; Liu, Y.; Liang, B.; Liang, Z.; Zhang, Y. Research on Microstructure and Cracking Behavior of Al-6.2Zn-2Mg-xSc-xZr Alloy Fabricated by Selective Laser Melting. Crystals 2022, 12, 1500. https://doi.org/10.3390/cryst12101500

Huang Shao, Tingting Liu, Kai Zhang, Changdong Zhang, Shuxin Jiang,Zhiwei Xiong & Wenhe Liao (2020): Preparation of Ti3AlC2 matrix composites by selective laser melting combined with pressureless sintering, Advances in Applied Ceramics, https://doi.org/10.1080/17436753.2019.1707411

Ghosh, S.K.; Saha, P.; Kishore, S. Influence of Size and Volume Fraction of SiC Particulates On Properties of Ex Situ Reinforced Al–4.5Cu–3Mg Metal Matrix Composite Prepared by Direct Metal Laser Sintering Process. Materials Science and Engineering: A 2010, 527 (18-19), 4694-4701. https://doi.org/10.1016/j.msea.2010.03.108

Xie, J.; Chen, Y.; Yin, L.; Zhang, T.; Wang, S.; Wang, L. Microstructure and Mechanical Properties of Ultrasonic Spot Welding TiNi/Ti6Al4V Dissimilar Materials Using Pure Al Coating. Journal of Manufacturing Processes 2021, 64, 473-480. https://doi.org/10.1016/j.jmapro.2021.02.009

Point 2) Please provide a quantitative analysis of void size/size distribution.

Response 2: We are sorry that we did not carefully check the data. Thank you for pointing out this problem to us. We have recalculated the pore size and drawn the pore size distribution in Supplementary Figure 4.

Point 3) How many times have you performed mechanical tests?

Response to 3: Thank you for your reminding, we add more details of mechanical property test in Page 6, Paragraph 2.3, line 203, the added sentences were “and the room temperature tensile test was carried out by the Quasar10 tensile testing machine. The test standard was GB/T 228.1-2010. Each valid data was repeated at least three times.

Point 4) Authors have mentioned “Due to the poor interfacial bonding between these part of the surface and the aluminum matrix, Mg (Zn1.5Cu0.5) phase or Al3(Sc,Zr) phase may be formed as the interphase to bond the matrix. The reason is that their lattice parameter at specific crystal plane is between TiB2 and aluminum matrix”.  I really do not think it is a good idea to talk about probabilities in conclusion. I suggest bring definite and certain findings in the conclusion. You can move this part to the discussion.

Response4: Thanks for your advice, The Mg (Zn1.5Cu0.5) phase acting as an intermediate phase combined with Al and TiB2 particles has been discussed systematically in published papers, so a more confident statement was adopted in our conclusion. The Al3(Sc,Zr) phase needs more experiments and data support, so we put its discussion on Page 19, Paragraph 1 and 2.

Ma, Y.; Addad, A.; Ji, G.; Zhang, M.; Lefebvre, W.; Chen, Z.; Ji, V. Atomic-Scale Investigation of the Interface Precipitation in a TiB2 Nanoparticles Reinforced Al–Zn–Mg–Cu Matrix Composite. Acta Mater. 2020, 185, 287-299. https://doi.org/10.1016/j.actamat.2019.11.068

Ma, Y.; Chen, H.; Zhang, M.; Addad, A.; Kong, Y.; Lezaack, M.B.; Gan, W.; Chen, Z.; Ji, G. Break through the Strength-Ductility Trade-Off Dilemma in Aluminum Matrix Composites Via Precipitation-Assisted Interface Tailoring. Acta Mater. 2023, 242, https://doi.org/118470.10.1016/j.actamat.2022.118470

Point 5) Again in paragraph 4: “The main reason for the increase may be the increase of densification due to the increase of powder laser absorption rate”. I do not like this uncertainty in the conclusion.

Response 5: Thanks for your advice, combining the previous studies with results of this experiment, we believe this conclusion can be supported, so we adopt a more confident statement in our conclusion. However, we also need to do more experiments in the future to further support this conclusion.

Li, X.P.; Ji, G.; Chen, Z.; Addad, A.; Wu, Y.; Wang, H.W.; Vleugels, J.; Van Humbeeck, J.; Kruth, J.P. Selective Laser Melting of nano-TiB2 Decorated AlSi10Mg Alloy with High Fracture Strength and Ductility. Acta Mater. 2017, 129, 183-193. https://doi.org/10.1016/j.actamat.2017.02.062

Gu, D.; Yang, Y.; Xi, L.; Yang, J.; Xia, M. Laser Absorption Behavior of Randomly Packed Powder-Bed During Selective Laser Melting of SiC and TiB2 Reinforced Al Matrix Composites. Optics & Laser Technology 2019, 119, 105600. https://doi.org/10.1016/j.optlastec.2019.105600

Point 6) Is 6% increase really a meaningful increase? Could that be a scattering in data?

Response 6: Your question is quite reasonable. However, according to our error analysis of mechanical properties, the properties of matrix AlZnMgCu(Sc,Zr) samples fabricated by SLM are very stable, with error less than 2%, as shown in Table 3. Although not much, we have reason to believe that the performance improvement of 6% is meaningful. Low mechanical performance improvement may be due to the low addition amount of reinforcement particles, only 3%. Later, we will introduce a larger addition or a finer particle in AlZnMgCu(Sc,Zr) for SLM forming, perhaps we can get some more mechanical improvement.

Point 7) The porosity is a very key issue in your paper. Yet, the discussion on the implications of porosity for the mechanical properties of your samples is very limited. I recommend you consult the following paper and add some discussion using this paper:

-        Statistical effects of pore features on mechanical properties and fracture behaviors of heterogeneous random porous materials by phase-field modeling. International Journal of Solids and Structures, 264, 2023, 112098. doi: https://doi.org/10.1016/j.ijsolstr.2022.112098

Response 7: Thanks for your valuable advice, we have supplemented the effect of pore size and shape on material properties in Paragraph 3.1 line 253, the added sentences were as below.

The size and shape of pores have great influence on Young's modulus, UTS and fracture behavior. There are more concentration points around large pores and irregular pores, so cracks are more likely to start and propagate nearby. The rounder the shape of pore, the more uniform the stress distribution nearby, and the higher the Young's modulus and UTS of the material [34].

Point 8) A revision of English is highly recommended

Response 8: We are very sorry that we have made many mistakes in English, which has caused misunderstanding. We rechecked and revised the manuscript. we hope it can get your approval.

We tried our best to improve the manuscript and made some changes in the manuscript. These changes will not influence the content and framework of the paper.

We appreciate for your warm work earnestly, and hope that the correction will meet with approval.

We look forward to hearing from you regarding our submission. We would be glad to respond to any further questions and comments that you may have.

Once again, thank you very much for your comments and suggestion. 

Best regards,

Mr. Peng yin

Round 2

Reviewer 1 Report

The keyword "Heterostructure with Columnar grain and Ultra-fine grain" is too long. Three shorter keywords phrase "Heterostructure", "Columnar grain" and "Ultra-fine grain" are possible to be presented separately instead of one long phrase.

Author Response

Further comments and suggestions: The keyword "Heterostructure with Columnar grain and Ultra-fine grain" is too long. Three shorter keywords phrase "Heterostructure", "Columnar grain" and "Ultra-fine grain" are possible to be presented separately instead of one long phrase.

Thanks again for your further advice, we use "Heterostructure" only as it can generalize these specific features, so "Columnar grain" and "Ultra-fine grain" are removed. In addition, the “Fracture behavior” is discussed in manuscript, so it is added. The final Keywords we are gona use are Selective laser melting, TiB2/AlZnMgCu(Sc,Zr) composites, Heterostructure, Al3(Sc,Zr), Fracture behavior
